# THz Spectroscopy as a Versatile Tool for Filler Distribution Diagnostics in Polymer Nanocomposites

**DOI:** 10.3390/polym12123037

**Published:** 2020-12-18

**Authors:** Gleb Gorokhov, Dzmitry Bychanok, Igor Gayduchenko, Yuriy Rogov, Elena Zhukova, Sergei Zhukov, Lenar Kadyrov, Georgy Fedorov, Evgeni Ivanov, Rumiana Kotsilkova, Jan Macutkevic, Polina Kuzhir

**Affiliations:** 1Institute for Nuclear Problems of Belarusian State University, Bobruiskaya Str., 11, 220006 Minsk, Belarus; dzmitrybychanok@ya.ru (D.B.); polina.kuzhir@gmail.com (P.K.); 2Physics Faculty, Vilnius University, Sauletekio 9, LT-10222 Vilnius, Lithuania; jan.macutkevic@gmail.com; 3Radiophysics Department, Tomsk State University, Lenin Avenue 36, 634050 Tomsk, Russia; 4Moscow Pedagogical State University, Malaya Pirogovskaya Str., 1/1, 119991 Moscow, Russia; igorandg@gmail.com; 5Moscow Institute of Physics and Technology, Institutskiy per., 9, 141701 Dolgoprudny, Russia; rogov.iup@phystech.edu (Y.R.); zhukovaelenka@gmail.com (E.Z.); zhukov@phystech.org (S.Z.); kadyrov@phystech.edu (L.K.); gefedorov@mail.ru (G.F.); 6OLEM, Institute of Mechanics Bulgarian Academy of Sciences, Acad. G. Bonchev Str., Bl. 4, 1113 Sofia, Bulgaria; ivanov_evgeni@yahoo.com (E.I.); kotsilkova@yahoo.com (R.K.); 7Research and Development of Nanomaterials and Nanotechnologies (NanoTech Lab Ltd.), Acad. G. Bonchev Str. Block 4, 1113 Sofia, Bulgaria; 8Institute of Photonics, University of Eastern Finland, Yliopistokatu 7, FI-80101 Joensuu, Finland

**Keywords:** nanocomposites, THz spectroscopy, percolation threshold

## Abstract

Polymer composites containing nanocarbon fillers are under intensive investigation worldwide due to their remarkable electromagnetic properties distinguished not only by components as such, but the distribution and interaction of the fillers inside the polymer matrix. The theory herein reveals that a particular effect connected with the homogeneity of a composite manifests itself in the terahertz range. Transmission time-domain terahertz spectroscopy was applied to the investigation of nanocomposites obtained by co-extrusion of PLA polymer with additions of graphene nanoplatelets and multi-walled carbon nanotubes. The THz peak of permittivity’s imaginary part predicted by the applied model was experimentally shown for GNP-containing composites both below and above the percolation threshold. The physical nature of the peak was explained by the impact on filler particles excluded from the percolation network due to the peculiarities of filler distribution. Terahertz spectroscopy as a versatile instrument of filler distribution diagnostics is discussed.

## 1. Introduction

Functional materials based on micro- and nanocarbon structures (single and multi-walled nanotubes, graphene flakes, carbon dots, nanohorns, etc.) are of great interest for electromagnetic (EM) applications alongside composites [1,2,3,4,5,6,7] and structures [8,9,10,11,12] on their basis. The simplest way to drastically change the mechanical [13,14,15,16] and EM [17,18,19,20] properties of a dielectric material (such as a polymer, ceramic or glass) is to fill it with conductive particles of microscopic/nanoscale size. Possessing the highest possible aspect ratio among all nanocarbon fillers, nanotubes are the first candidate to create a conductive network inside a composite, i.e., reach the electric percolation threshold at the smallest possible concentration [2,21,22,23]. Another widely-used nanocarbon filler, namely, graphene nanoplatelets, usually have the percolation threshold of 3.5–10% [24,25,26,27,28]. That is several orders higher than the 0.002% percolation threshold possible for composites based on single-walled nanotubes [2,23,29].

Multi-filler composites containing different types of fillers (including non-carbon) tend to form heterogeneous percolation networks [30,31,32]. This fact allows making composite materials without expensive fillers through the introduction of low-cost ones into the hybrid percolation network [33,34], or even enhancing the electromagnetic properties due to the synergistic effect [35,36,37]. Recently, bi-filler composites containing graphene nanoplatelets and carbon nanotubes have attracted great interest due to the possible synergistic effects in their mechanical, thermal and electric properties [38,39,40,41]. Besides the filler morphology, the percolation threshold value strongly depends on the composite preparation technology, the filler dispersion in the dielectric phase (matrix) and the filler–polymer interfacial interactions. In previous studies on the herein investigated nanocomposites, the electrical percolation thresholds of 0.5% for MWCNT/PLA, 6% for GNP/PLA and 3% for the bi-filler composite GNP/MWCNT/PLA were reported [42]. In the framework of the current research, we speak only about the electric percolation, i.e., the interconnected conductive network formation of filler particles, where the dielectric matrix plays a direct role only in the electron tunneling [43]. However, the characteristics of polymer and/or composite preparation technology may significantly affect the distribution of filler particles.

The dielectric properties of composites above the percolation threshold are mainly governed by electrical transport and Maxwell–Wagner relaxation. However, at higher frequencies, the contributions of big percolating clusters to the Maxwell–Wagner relaxation and the electrical conductivity become less important, so that in the terahertz frequency range it is possible to separate the contributions of individual nanoparticles’ polarizations from the total dielectric permittivity [44]. The most commonly used non-destructive method of composite system characterization is low-frequency impedance spectroscopy, which allows one to easily establish percolation thresholds. However, the dispersion of nanoparticles may be investigated by other direct methods, such as transmission and scanning electron microscopy, and indirect methods (noise spectroscopy [45], magnetic susceptibility measurements [46], etc.) [3,47,48].

Time-domain terahertz spectroscopy is a non-destructive diagnostic method widely applied for the characterization of polymer-based nanocomposites’ [49,50,51] and thin films’ [52,53,54] electromagnetic properties in the 100 GHz–3 THz frequency range without any predefined assumptions. Being in between microwave and optical infra-red frequency regions, terahertz radiation easily transmits through most dielectric materials, while the metals and certain dielectrics, such as water, are non-transparent at the mentioned frequencies. Since the inhomogeneities’ dimensions are orders less than the wavelength, spectral parameters of nanocomposites may be investigated in the framework of effective medium theory [55,56].

It is worth noting that the percolation threshold is usually considered as a filler concentration at which the percolation network appears. However, due to the imperfections of filler distribution, the actual number of particles involved in the percolation network is lower than the threshold. The current investigation is aimed at distinguishing the impacts of filler particles excluded from the percolation network on the terahertz electromagnetic properties of a composite. Additionally, the possibilities to tune the frequency dispersion of permittivity in THz range by variation of filler contents in composites based on GNP, MWCNT and their mixtures are discussed.

## 2. Materials

### 2.1. Composites Fabrication

The polymer matrix used in this study was Ingeo™ Biopolymer PLA-3D850 (Nature Works, Minnetonka, MN, USA). The nanofillers used were: commercially available industrial graphene nanoplates, TNIGNP (supplied by TimeNano, Chengdu, China), with 90 wt.% purity; number of layers < 30; thickness < 30 nm; diameter/median size 5–7 μm; aspect ratio: ∼230/165; and volume resistivity < 0.15 Ohm/cm. Industrial grade OH-functionalized carbon nanotubes (multi-walled carbon nanotubes; MWCNTs; TimeNano, produced by CVD method) with 95 wt.% purity; 2.48 % OH^−^ content; size (outer D = 10–30 nm, length = 10–30 μm); aspect ratio: ∼1000; and 100 S/m electric conductivity. A high amount of GNP impurities has an insufficient impact on the PLA composite processing due to the slip effect between GNPs in the PLA matrix during the shear flow [57]. Impurities mostly consist of amorphous carbon; thus their impacts on the electromagnetic properties may be neglected.

Twin-screw extruder (COLLIN Teach-Line ZK25T) was used to prepare nanocomposites at temperatures of 170–180 °C and screw speed 40 rpm in two runs. The mono- and bi-filler nanocomposite hybrids were processed using the melt extrusion method, which includes the preparation of master batches, and further dilution. It was previously shown that both GNP and MWCNT addition significantly suppress the thermal degradation and the aging of polymer nanocomposites (accelerated by humidity uptake, UV light, etc.), compared to the neat PLA, due to the nucleation and the barrier effects of nanofillers [58].

The mono-filler composites (PLA/MWCNT and PLA/GNP) with 1.5 wt.%, 3 wt.% and 6 wt.% filler contents, and the bi-filler composites (PLA/MWCNT/GNP) with 3 wt.% and 6 wt.% total filler content (combining GNP and MWCNT in different proportions) were prepared. Thin films of nanocomposites were obtained by hot pressing at 180 °C and pressure of 1 bar. Before pressing the test samples, composite pellets were dried in a vacuum oven at 80 °C for 4 hours in order to minimize the humidity uptake. The applied temperature range was significantly lower than the PLA thermal degradation onset (230 °C). The physicochemical characteristics of the PLA-based carbon nanocomposites under investigation are given in Table 1. Composites’ crystallinity was 30%.

### 2.2. Sample Characterization

According to the low-frequency measurements described before (see Figure 4 in [40]), the percolation threshold for MWCNTs lies between 1.5 and 3 wt.%, while GNP-based composite experiences percolation between 3 and 6 wt.%. Therefore, the set of samples under investigation contains both pre- and post-percolated composites.

The quality of nanoparticles and their distribution in the polymer matrix were examined by means of a Raman spectrometer combined with the confocal microscope (Nanofinder High End, Tokyo Instruments, Belarus-Japan). Raman spectra were obtained using 100X objective with NA = 0.95 and the spot size on the sample surface was 0.75 μm. The excitation source was a 473 nm laser.

It is known that the typical Raman spectra of carbon nanomaterials, including MWCNTs, GNP, graphene, etc., are dominated by three characteristic peaks centered at ∼1360 cm^−1^, ∼1580–1600 cm^−1^ and ∼2600–2700 cm^−1^ (usually referred to as D, G and 2D modes, respectively) [59,60,61]. On the other hand, the strongest features of pure PLA are located in the vicinity of 3000 cm^−1^. Comparing the Raman spectra collected from different points on the surface of the composite containing 1.5% GNP allows one to figure out GNP particles, points 1 and 2 in Figure 1b, correspond to the filler-free area and GNP, respectively. The contrast difference between GNP particles and pure polymer Figure 1a indicates the particle distribution in the polymer matrix. It is worth noting that pure PLA is optically transparent; hence, it is possible to observe GNP particles under the sample surface. However, the addition of a small MWCNT amount makes the PLA matrix non-transparent. As a result, in the bi-filler composites, GNP particles can be identified by optical microscope (Figure 1c) only on the sample surface. As can be seen from the Raman spectra (Figure 1d), in the bi-filler composites nanotubes are uniformly distributed in the polymer matrix. However, at different points in the sample, the proportions of MWCNTs and PLA impacted the Raman spectrum differently (see points 1 and 3 in Figure 1d). Raman spectroscopy allows distinguishing even a small number of MWCNTs which are present on the surfaces of GNP particles (the observed D mode indicates CNTs at point 2).

Bright-field transmission electron microscopy (TEM) analysis was performed using a FEI TECNAI G12 Spirit-Twin (LaB6 source) instrument equipped with an FEI Eagle-4k CCD camera operating with an acceleration voltage of 120 kV. The analysis was performed on sections obtained at room temperature by using a Leica EM UC6/FC6 ultramicrotome. The sections were placed on 400 mesh copper grids. According to the transmission electron microscopy (TEM) of obtained composites, the method applied allows obtaining a mostly uniform distribution of both filler particle types. Figure 2a–c proves that GNPs kept their dimensions after the melt extrusion processing.

The TEM images of MWCNTs in mono- (Figure 2d,e) and bi-filler (Figure 2f) composites show that in both cases, the nanotubes are well-distributed. Due to the significantly higher aspect ratio, they tend to form percolation networks, even at the smallest investigated concentration (compare the number of percolating clusters in Figure 2d,e). However, independently of concentrations and the percolation existence, all prepared composites contain a certain number of insulated filler particles. Summarizing the above, both nanocarbon fillers possess good dispersion in the PLA matrix.

## 3. Experimental: Time-Domain THz Transmission Measurements

In order to perform the THz measurements, the thicknesses of samples were reduced to 200–300 μm by precise hand polishing with a diamond paste which was removed with isopropanol and water. During the processing, composites were kept near room temperature to prevent heat-induced variation of their structure. All films were studied by transmission microscopy, which requires very thin samples. The terahertz–subterahertz spectra (5–60 cm^−1^) of the films were measured by means of commercially available pulsed time-domain spectrometer (TDS) TERA K15 (Menlo Systems). In a TDS spectrometer, the sample is exposed to a picosecond pulse that contains frequency components in a wide range up to several terahertz. The position and amplitude of the pulse are detected when the measurement channel is empty and blocked by the sample. The difference in time between the two peaks is a measure of the radiation delay caused by the sample, and a peak’s amplitude gives the measure of radiation absorption in the sample. The transition from the time domain to the frequency domain was implemented using the Fourier transformation, and resulted in the spectra of transmission coefficient amplitude and phase. This data were sufficient to determine the real and imaginary parts of the complex conductivity and dielectric permittivity of the investigated sample. The latter, written as $\varepsilon =\varepsilon^{\prime }-i\varepsilon^{\prime \prime }$, consists of two parts. The real part of permittivity, $\varepsilon^{\prime }$, corresponds to the energy storage capacity of a material, and the imaginary part (or sometimes, the loss factor), $\varepsilon^{\prime \prime }$, is a measure of all dissipation effects in a material (see Section 1.2.2.1 and 1.3.1 in [62] for further reference).

Experimental values of complex permittivity are shown in Figure 3. The imaginary part of permittivity (further referred to as $\varepsilon^{\prime \prime }$) demonstrates a peak for most of the investigated samples containing GNP filler. On the contrary, the MWCNT-based monofiller composites and the bi-filler composites with 1.5% GNP and 4.5% MWCNT demonstrated a monotonous decrease with frequency growth, which is typical for percolated systems. Finally, the non-percolated composite containing 1.5 wt.% MWCNT did not demonstrate the $\varepsilon^{\prime \prime }$ peak in the investigated frequency range.

Another feature of the electromagnetic response is very close values of the real part of permittivity for all three bi-filler composites containing 6 wt.% nanocarbon, accompanied by significant variation in the imaginary part. In Section 5 we will try to clarify the physics origin of these peculiarities inherent for two different kinds of complex permittivity behavior, characteristic for percolated and non-percolated composites, and whether the filler particles’ distribution in the polymer could play a significant role.

## 4. Modeling: MG Effective Medium Theory

One of the well-known approaches to describe the electromagnetic properties of a micro-or nanocomposite material is to apply the effective medium theory. In the case when the filler concentration is lower than the percolation threshold, i.e., filler particles are not interacting, the Maxwell Garnett (MG) effective medium theory is applied [26,63]. This approach assumes the filler particles as ellipsoids with semiaxes a,b,c, and the following polarizability:(1)$$\alpha_{i}\left(\nu ,\sigma \right)=\frac{4\pi abc}{3}\frac{\varepsilon_{m}\left(1-\frac{i\sigma }{2\pi \nu \varepsilon_{0}}-\varepsilon_{m}\right)}{\varepsilon_{m}+N_{i}\left(1-\frac{i\sigma }{2\pi \nu \varepsilon_{0}}-\varepsilon_{m}\right)},$$
where σ is the ellipsoid conductivity. $\varepsilon_{0}$ is vacuum permittivity; $N_{i}$ is the depolarization factor in direction i=a,b,c which is given as:(2)$$N_{i}=\frac{1}{abc}\int_{0}^{\infty }\frac{ds}{\left(s+i^{2}\right)\sqrt{\left(s+a^{2}\right)\left(s+b^{2}\right)\left(s+c^{2}\right)}}.$$

Considering the above, the effective dielectric permittivity is:(3)$$\varepsilon_{eff}=\varepsilon_{m}+\frac{1/3\sum_{i=a,b,c}n\alpha_{i}/V}{1-1/3\sum_{i=a,b,c}\frac{N_{i}n\alpha_{i}/V}{\varepsilon_{m}}},$$
where $\varepsilon_{m}$ is the dielectric matrix permittivity; *n* is the volume concentration of filler.

It is possible to show (see the example in Figure 4) that the imaginary part of polarizability calculated by Equation (Equation 1) experiences a maximum at the critical frequency (Equation (Equation 4)). Therefore, the imaginary part of permittivity will experience the same peak.
(4)$$\nu_{c}=\frac{N_{i}\sigma }{2\pi \varepsilon_{0}\left(\varepsilon_{m}-N_{i}\varepsilon_{m}+N_{i}\right)}.$$

## 5. Results and Discussion

### 5.1. MG Approximation of GNP-Based Composites Permittivity

Typical values for the percolation threshold for GNPs in polymer composites are usually not lower than 3.5% and strongly depend on the GNP aspect ratio, the polymer type and the quality of particles’ distribution [25,27]. When the wavelength is orders larger than GNPs lateral dimensions, it is possible to apply the Maxwell Garnett effective medium theory [44]. Let us assume that the GNP particles are uniform ellipsoids with a=b≫c semiaxes and consider their density equal to graphite (2.2 g/cm^3^). The typical aspect ratio (AR) for graphene nanoplatelets lies in the range of 100–1000. An acceptable fitting of dielectric permittivity of composites containing 1.5 and 3% GNP (Figure 5) was acquired with σ = 20,000 S/m and AR ∼ 200, which is in good agreement with the aspect ratio of used GNP particles given in Section 2.1. While the approximation parameters for composites with 1.5 and 3% GNP concentration are similar, the sample containing 6 % filler requires another set of parameters to be used for approximation. While the conductivity of ellipsoids remains unchanged, the aspect ratio has to be increased up to nearly 300.

The fact that the Maxwell Garnett approximation is still applicable for 6 % GNP may indicate that in spite of the concentration being higher than previously reported percolation thresholds for GNPs [40], the impact of insulated filler particles still prevails. In the next subsection, we will try to explain the low-frequency shift of $\varepsilon^{\prime \prime }$ maximum.

### 5.2. Imaginary Permittivity Peak in Percolated Mixed-Filler Composites

In the previous section, it was shown that the composites with GNP concentrations below the percolation threshold demonstrate the imaginary permittivity maximum predicted in Section 4. However, the experimental data for composites containing a mixture of fillers show a similar behavior, even with the existence of percolation (Figure 3).

In a perfect case, every particle of both fillers is expected to be involved in a heterogeneous percolation network, making the Maxwell Garnett approach inapplicable. However, the real composite system always contains a certain amount of filler particles excluded from the percolation network formation. At low frequencies, the response of such insulated particles is hidden by the percolation network impact, but as in the terahertz region, it is possible to distinguish their impact due to electromagnetic coupling.

In Figure 6a the complex permittivities of composites containing 1.5% GNP, 1.5% MWCNT and their mixture are depicted. Both parts of this mixture have concentrations below the percolation threshold. It can be seen that despite the imaginary part of MWCNT-based composite, the permittivity has no peculiarities, such as peaks in the investigated range; the imaginary part of the GNP-MWCNT bi-filler composite’s permittivity shifts towards lower frequencies in comparison with peak inherent for the GNP-based filler.

A similar situation was observed for a mixture of 3% MWCNT and 3 wt.% GNP composites (Figure 6b), a set of mixtures with 6 wt.% nanocarbon (Figure 3) in which the nanotube content is above the percolation threshold. In both cases, it is possible to conceive the dispersion of bi-filler composite’s ε as a superposition of MWCNT (monotonous decrease of both real and imaginary parts of permittivity) and GNP (peak of $\varepsilon^{\prime \prime }$) curves. Considering Equations (Equation 1)–(Equation 3), the $\varepsilon^{\prime \prime }$ shift can be explained by the growth of the average aspect ratio and/or the conductivity decrease (see Figure 2 in [44]). Due to the higher aspect ratio (∼1000), non-percolated nanotubes will give a $\varepsilon^{\prime \prime }$ peak at lower frequencies in comparison with GNPs. That means the GNPs and MWCNTs may be both involved in the percolation network and excluded from it.

As soon as the Maxwell Garnett approach does not require the dielectric nature of the matrix, it is possible to substitute the $\varepsilon_{m}$ in Equation (Equation 3) with the permittivity of a perfect percolated composite (every particle of which is involved in percolation network). Then, the effective permittivity of the mixed-filler composite may be introduced as following:(5)$$\varepsilon_{effmix}=\varepsilon_{MWCNT}+\Delta \varepsilon_{cp}+\Delta \varepsilon_{np}$$
where the $\varepsilon_{MWCNT}$ term is the permittivity of MWCNT-based composite, $\Delta \varepsilon_{cp}$ is a “synergistic” impact of cross-MWCNT-GNP percolation and $\Delta \varepsilon_{np}$ is an impact on the dielectric permittivity made by non-percolated GNP particles. The expression for $\Delta \varepsilon_{np}$ is similar to the second term on the right side of Equation (Equation 3).

Results of rough $\Delta \varepsilon_{cp}$ evaluation (considering $\Delta \varepsilon_{np}=\varepsilon_{GNP}-\varepsilon_{PLA}$) for mixed-filler composites containing 1.5 % MWCNT + 1.5% GNP and 3 % MWCNT + 3 % GNP are presented in the Figure 7.

It can be seen that the $\Delta \varepsilon_{cp}$ frequency dispersion is typical for composites whose filler particles are mostly involved in the percolation network. However, the non-percolated nanotubes’ impact value remains hidden in the total $\varepsilon_{MWCNT}$. The $\varepsilon^{\prime \prime }$ shift characteristic to the bi-filler composite (Figure 6) may be considered as indirect proof that a certain number of MWCNTs are excluded from the percolation network.

## 6. Conclusions

The impact of nanofiller particles excluded from the percolation network was demonstrated by means of transmission time-domain terahertz spectroscopy. The theoretically predicted peak of imaginary permittivity was observed on PLA-based nanocomposites filled with GNP, MWCNT and their mixtures both below and above the percolation threshold. Terahertz spectroscopy was demonstrated as a sensitive tool for the estimation of filler distribution character. For instance, the agreement between the experimental value of complex permittivity obtained for mono-filler GNP-based composites and Maxwell Garnett effective medium theory approximation indicates the mostly uniform distribution of insulated graphene nanoplatelets inside the polymer matrix.

The strong variation of the imaginary part of the permittivity dispersion followed by the relatively weak impact to its real part was experimentally shown for the set of bi-filler composites containing 6 wt.% of nanocarbon fillers in total. The variation of GNP fraction in the ternary composite resulted in a change of imaginary permittivity peak frequency, and the real part of permittivity remained mostly unchanged, giving the possibility of electromagnetic property tuning. A simple variation of fillers proportion during the production of the composites by co-extrusion method allows one to precisely control the position of the $\varepsilon^{\prime \prime }$ peak.

Finally, the combination of different nanocarbon fillers allows precise modification of the EM properties of a composite filled with relatively cheap GNPs by addition of CNTs, thereby obtaining a material with high THz performance.

## Figures and Tables

**Figure 1 polymers-12-03037-f001:**
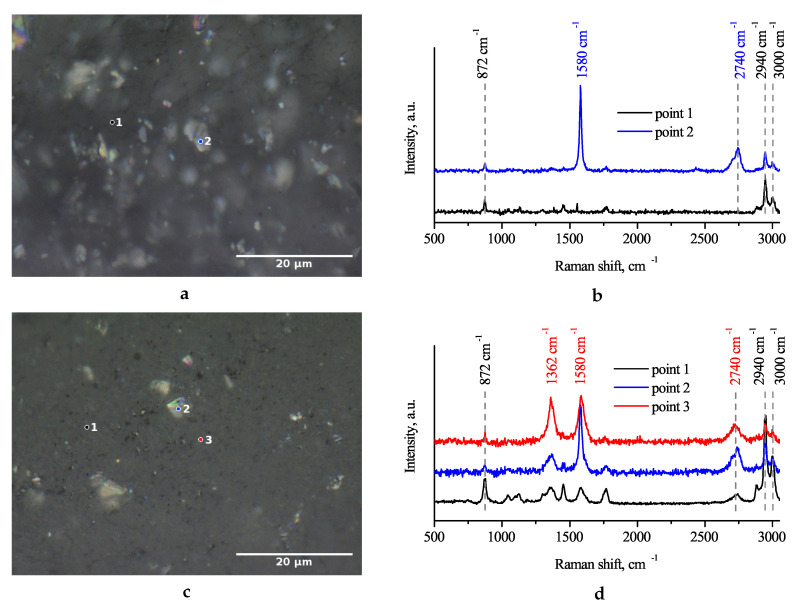
Optical microscopy image (**a**) and Raman spectra (**b**) of PLA composite containing 1.5% GNP. Due to the neat PLA transparency, it is possible to see GNPs under the surface of a composite. Optical microscopy image (**c**) and Raman spectra (**d**) of PLA composite containing the mixture of 3% GNP and 3% MWCNT. Raman spectrum collected at point 2 combines the characteristic peaks of PLA, GNPs, and MWCNTs.

**Figure 2 polymers-12-03037-f002:**
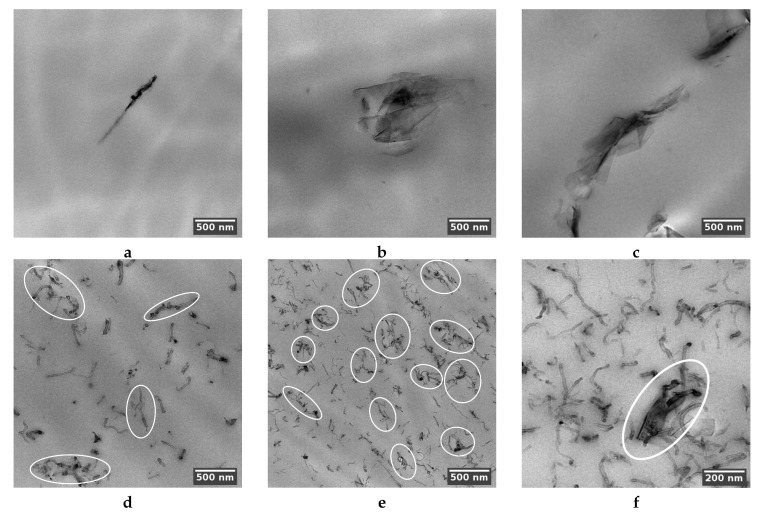
TEM images of insulated filler particles in PLA composites with 1.5 (**a**), 3 (**b**), and 6 wt.% (**c**) GNP; insulated particles and percolating clusters (enclosed in ovals) in composites containing 1.5 (**d**) and 3 wt.% (**e**) MWCNT; the mixture of 3% GNP and 3% MWCNT (**f**). A graphene nanoplatelet involved in percolation contact with nanotubes is enclosed in an oval.

**Figure 3 polymers-12-03037-f003:**
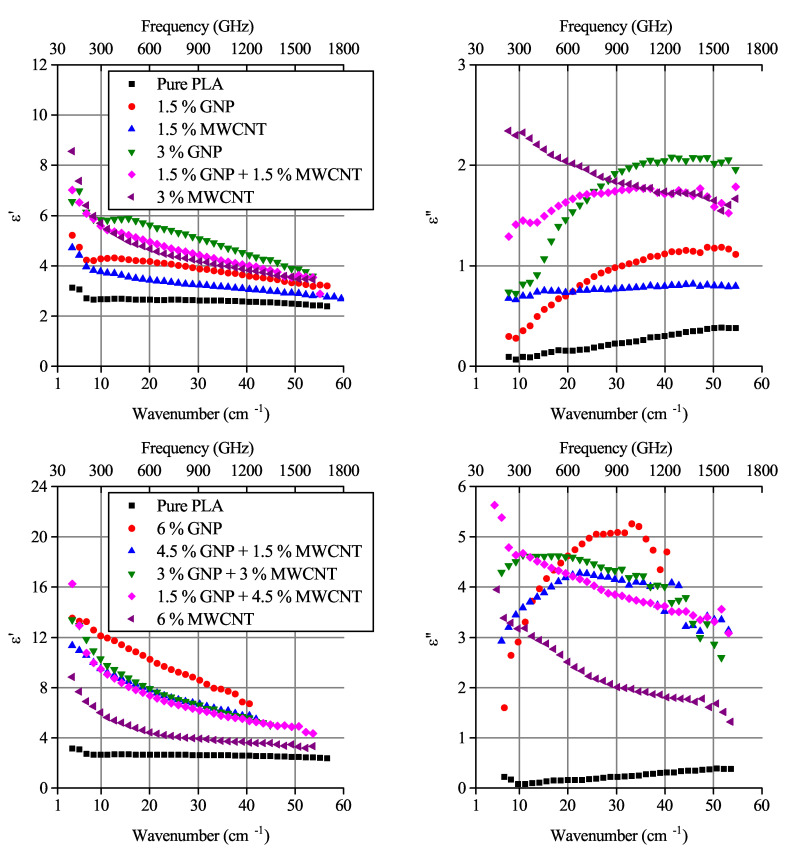
Complex permittivity of composites containing 3 (**top**) and 6 (**bottom**) wt.% nanocarbon fillers. MWCNT concentrations are above the percolation threshold.

**Figure 4 polymers-12-03037-f004:**
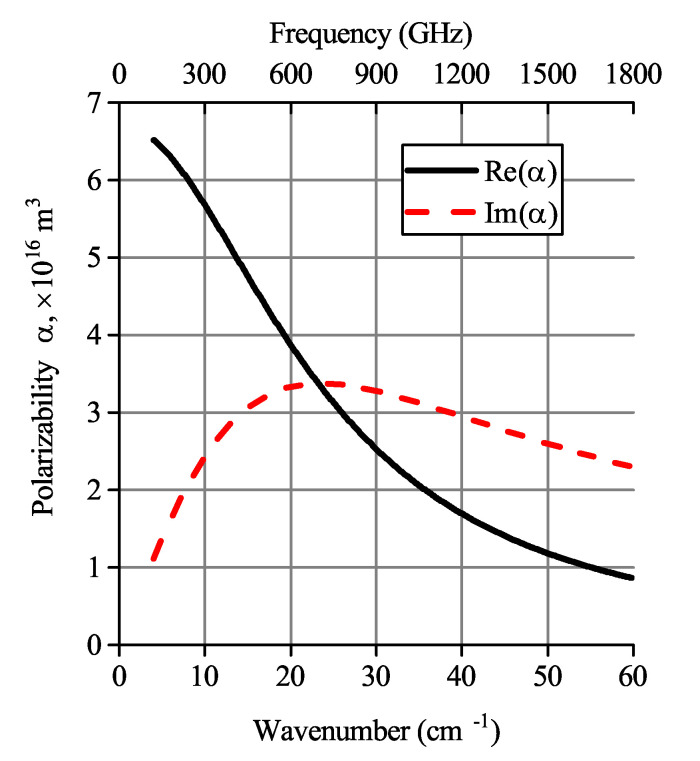
Frequency dispersion of the longitudinal polarizability calculated for oblate ellipsoid with *a* = *b* = 5 μm, *c* = 25 nm (i.e., AR = 200), σ=10,000 S/m and $\varepsilon_{m}=2.5-0.1i$.

**Figure 5 polymers-12-03037-f005:**
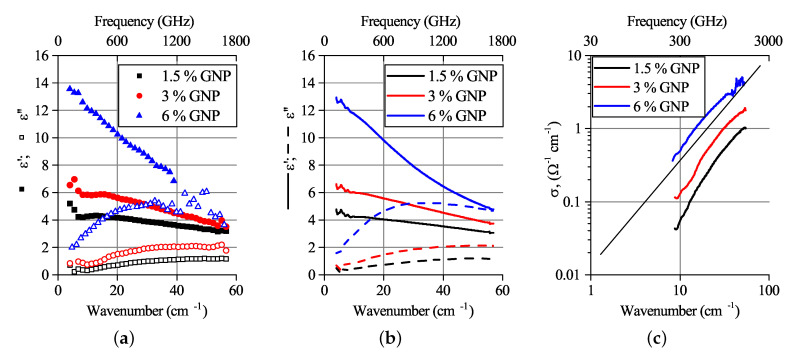
(**a**) Complex permittivity of composites containing 1.5, 3 and 6 wt.% GNP; (**b**) Maxwell Garnett approximation of the experiment with the following fitting parameters: σ = 20,000 S/m, and AR ∼ 197, 208 and 291, respectively; (**c**) conductivity values of GNP-based composites having similar slopes.

**Figure 6 polymers-12-03037-f006:**
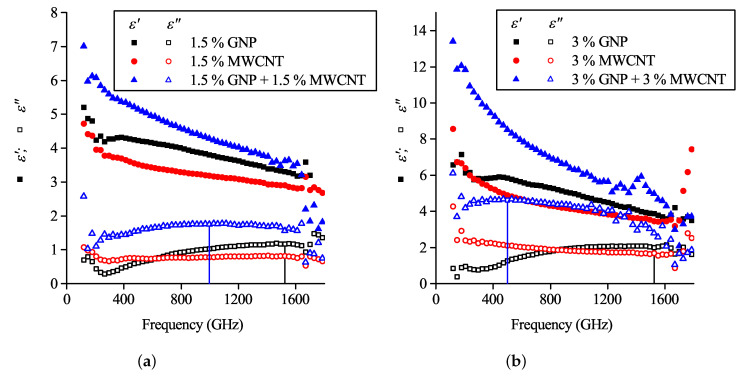
Complex permittivities of composites containing (**a**) 1.5% GNP, 1.5% MWCNT and their mixture; (**b**) 3% GNP, 3% MWCNT and their mixture. Peak values of $\varepsilon^{\prime \prime }$ for GNP and GNP + MWCNT composites are indicated by vertical lines.

**Figure 7 polymers-12-03037-f007:**
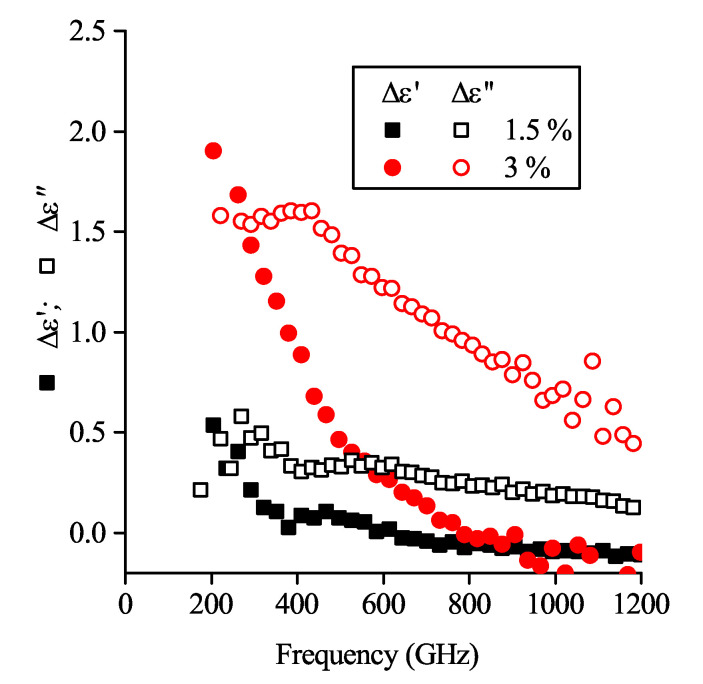
Impacts ($\Delta \varepsilon_{cp}$) of cross-filler percolation to the dielectric permittivity of composites containing 1.5% MWCNT + 1.5 % GNP and 3% MWCNT + 3% GNP (referred to as 1.5% and 3%, respectively). Both real and imaginary parts behave typically for a percolated composite.

**Table 1 polymers-12-03037-t001:** Physicochemical characteristics of polymer composites after hot pressing [43,58].

Parameter	Value
Glass transition temperature $T_{g}$, °C	65
Cold crystallization temperature *T_cc_*, °C	87–92
Melting temperature *T_m_* , °C	175
Melt crystallization temperature *T_g_*, °C	105
Crystallinity, %	30

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
