# Peer review of "THz Spectroscopy as a Versatile Tool for Filler Distribution Diagnostics in Polymer Nanocomposites"

_polymers, 2020, doi:10.3390/polym12123037_

Round 1

Reviewer 1 Report

The manuscript polymers-1004627 presents an experimental evaluation of the potential of THz spectroscopy, a very interesting Non-Destructive Testing method, to investigating the structure–property correlations in conducting polymer-graphene and/or carbon nanotube composites.

The general aim of the work is clearly drawing our attention. However, several key-points should be revised before considering the overall pertinence of the results and discussions.

General comment: The disproportionate / unjustified use of self-citations (50% of the references: [3]-[9], [19]-[24], [27]-[28]) in a field with a large validated and accessible database limits the scientific soundness of Introduction and Discussion parts and also allowed experimental unhappy choices that risk to compromising the pertinence of the experimental results.

I consider the manuscript needs major revision.

Author Response

Dear Reviewer, thanks for a lot of attention you've given to our manuscript.

Your questions, especially considering the composites preparation process, helped us to improve the quality of our results presentation. Our comments about your questions and our answers you can find in the attached PDF document. Most of those corrections were suitable for the manuscript. However, after the discussion, we decided to remain the structure simple and linear, because now we consider it more simple for reading.

On behalf of the research team,

Gleb Gorokhov

Institute for Nuclear Problems, Belarusian State University, Minsk, Belarus.

Reviewer 2 Report

The manuscript is recommended for publication after a minor revision.

(a) A thorough check in English is required for resubmission.

(b) The quality of some figures (3-7) should be improved. 

Author Response

Dear Reviewer, thanks for giving an attention to our manuscript.

We've performed a strict grammar check, so now, we hope, it is easier to read. Please, be aware, that we significantly improved certain parts of manuscript not only answering the Reviewers questions, but attempting to make it more clear.

The quality of figures was also increased. However, if you still have something to improve considering the figures, please, be more explicit. As we can judge, the pictures are readable enough.

On behalf of the research team,

Gleb Gorokhov

Institute for Nuclear Problems, Belarusian State University, Minsk, Belarus

Round 2

Reviewer 1 Report

The revised version of the manuscript polymers-1004627 is much clearer and precise in the description of experimental conditions of (i) compounding the polymer nanocomposites and (ii) polishing the samples and in presenting the general context of this work and the original part.

The discussion of results is also better presented, except for the TEM results.

As mentioned in my 1st review, the Figures 1a-c and Figure 2c are not supporting the text statements (lines 108-110): “The TEM images in high magnification are presented in order to demonstrate the interconnected nanofiller particles around the percolation threshold, as well as the presence of nanoparticle-free spaces in the dielectric polymer matrix.”

Or, the revised manuscript presents same Figures 1 a-c and Figure 2c as in the 1st version.

More precisely:

  • Figure 1 is not representative for the overall morphology of GNP-nanocomposites and does not support the text statement (lines 99-100): “TEM images in Figure 1 indicate that GNPs are mostly insulated, however at 6 wt. % concentration they tend to agglomerate (Figure 1, (c)).”

Figures 1(a)-(c) should offer for each GNP-nanocomposite: a more general view of the morphology (lower magnitude) + the current zoom image of isolated (packets of) GNPs.

In the present form, one can only suppose a heterogeneous composite morphology due to ineffective compounding conditions.

(GNP: thickness < 30 nm; diameter/median size 5 – 7 microns; aspect ratio: 230/165 --- MWCNTs size: outer D = 10 – 30 nm, length = 10 – 30 microns; aspect ratio: 1000)

  • In Figure 2c, due to the different scalebar as compared to 2a and 2b, it is difficult to identify the presence of GNP in addition to MWCNTs, especially because in 2b one can already see small MWCNT-packets that could look like the dark spot in Figure 2c, when passing from 500 nm to 200 nm scalebars.

In order to observe and understand the overall morphology of bi-nanocomposite, an image at 500 nm scalebars + the current image (or equivalent), both indicating with arrows the two different nanoparticles and their dispersion inside the polymer matrix, are necessary.

I need to insist on these points as they are crucial for correctly understanding and proving the relation with THz results.

Besides, the authors are pointing out explicitly the importance of TEM observations for correctly interpreting the experimental Tz results – lines 110-111: “Such structure visualization is of importance to relate to the THz results in this manuscript.”

My second observation points out the fact that the Introduction and the Discussion part are largely based on self-citations (in total 16/50), indicating a limited study of the current state-of-the-art on this topic. This remains a weak point of the manuscript, as the revised manuscript did not significantly change it.

I consider that at least the morphology presentation needs to be improved before considering the manuscript for publication.

Author Response

Dear Reviewer, thanks for attention you've given to our manuscript.

We are thankful for your valuable comments and questions. Your criticism about our TEM interpretation is reasonable, so we made additional investigation of composites structure by means of optical microscopy and Raman spectroscopy. Now, we hope, the microscopic characterization of investigated composites is explicit.

Please find full-text response in the attachment.

On behalf of the research team,

Gleb Gorokhov

Institute for Nuclear Problems, Belarusian State University, Minsk, Belarus
